# Isolation, identification, and whole-genome sequencing of high-yield protease bacteria from Daqu of ZhangGong Laojiu

Yanbo Liu[1,2,3,4], Junying Fu[1,3,4], Linlin Wang[1,3,4], Zhijun Zhao[1,3,4], Huihui Wang[1,3,4], Suna Han[2], Xiyu Sun[1,3,5], Chunmei Pan[1,3,4]*

**1** College of Food and Biological Engineering (Liquor College), Henan University of Animal Husbandry and Economy, Zhengzhou, Henan Province, China, **2** Postdoctoral Programme, Henan Yangshao Distillery Co., Ltd., Mianchi, Henan Province, China, **3** Henan Liquor Style Engineering Technology Research Center, Henan University of Animal Husbandry and Economy, Zhengzhou, Henan Province, China, **4** Zhengzhou Key Laboratory of Liquor Brewing Microbial Technology, Henan University of Animal Husbandry and Economy, Zhengzhou, Henan Province, China, **5** ZhangGongLaoJiu Wine Co. Ltd., Ningling, Henan Province, China

* sige518888@163.com

**Data Availability Statement:** All relevant data are within the manuscript and its Supporting Information files.

## Abstract

A total of 296 strains of protease-producing bacteria were isolated and purified from medium-temperature Daqu produced by ZhangGong LaoJiu Wine Co. Ltd. After calculating the ratio of transparent ring diameter to colony diameter and measuring the protease activities, a strain of high-yield protease bacteria, called DW-7, was screened out with a protease activity of 99.54 U/mL. Through morphological observation, 16S rDNA sequence analysis, and physiological and biochemical tests, the isolated bacteria DW-7 was determined to be *Bacillus velezensis*. In addition, whole-genome sequencing (WGS), using PacBio and the Illumina platform, was performed. Gene annotation was then conducted using the Clusters of Orthologous Groups (COG), Kyoto Encyclopedia of Genes and Genomes (KEGG), Non-Redundant Protein Sequence Database (NR), and Gene Ontology (GO) databases. The results showed that the genome of DW-7 was 3,942,829 bp long with a GC content of 46.45%. A total of 3,662 protein-encoding genes were predicted, with a total length of 3,402,822 bp. Additionally, 2,283; 2,796; and 2,127 genes were annotated in the COG, KEGG, and GO databases, respectively. A total of 196 high-yield protease genes were mainly enriched in the metabolism of alanine, aspartic acid, glutamate, glycine, serine, and threonine, as well as ABC transporter and transporter pathways.

## Introduction

Chinese Luzhou-flavored liquor is typically created via an oriental solid-state fermentation process. Liquor fermentation is a complex process that takes place in in an open environment and is coordinated by various microorganisms and enzymes. The microorganisms in liquor mainly come from the Daqu, the surrounding environment, pit mud, and yellow water [1]. Daqu is a unique saccharification starter for solid liquor brewing in China. It not only contains

**Funding:** This work supported by the Major Science and Technology Projects of Henan Province of China(181100211400), Key Technologies Research and Development Program of Henan Province of China(202102110130) , Scientific Research Foundation for Docotors of Henan University of Animal Husbandry and Economy(2018HNUAHEDF011) and Key Subject Projects of Henan University of Animal Husbandry and Economy(C3060020), all awarded to YL. The funders had no role in study design, data collection and analysis, decision to publish, or preparation of the manuscript.

**Competing interests:** The authors have declared that no competing interests exist.

important crude enzymes and complex microflora, but it is also a source of important biologically active substances, such as aroma precursors that determine flavor characteristics [2, 3]. The quality of Daqu directly affects the quality, yield, and style of the liquor product.

The microorganisms in Daqu mainly include bacteria, yeasts, molds, and a small number of actinomycetes. Bacteria are the vital microorganism in Daqu, the amount and quantity of which directly determines its quality. *Bacillus sp.* is the dominant flora in Daqu, which produces various enzymes that are closely linked to the formation of flavor substances during the fermentation process [4]. It has been reported that many flavor substances in liquor are derived from proteins, and the functions of those proteins and their enzymes are s significant for developing the flavor of liquor. For example, protease is an enzyme that hydrolyzes peptide bonds into amino acids or short peptides [5] and functions by maintaining enzyme activities under acidic, neutral, and alkaline conditions. Additionally, it is a metabolite of various microorganisms involved in the production of Daqu [6]. In the process of solid-state liquor fermentation, proteases in Daqu can promote the dissolution of raw materials, decompose proteins, produce amino acids, provide nitrogen sources, and flavor components, promote the growth of microorganisms, degrade yeast cells, improve the utilization of raw materials, cooperate in co-fermentation with other microbes, inhibit the production of fusel oil, and enhance the yield, quality, and aroma of the original liquor [7, 8].

Whole-genome sequencing (WGS) is a fast, low-cost, and highly efficient third-generation sequencing technology that can provide a complete bacterial genome sequence. The differences between species of the same genus can be better identified using WGS and by following gene annotations using online databases, such as GO, KEGG, COG, and NR. WGS has become a popular detection method and is widely used to identify microbial communities within intestinal flora, soil, and fungi [9, 10]. It is necessary for mining the core genome, identifying specific genes, and analyzing functional genomics, which ultimately contributes to the exploration of the diversity and biological characteristics of unknown microbial groups [11].

At present, domestic and foreign research mainly focuses on the physical and chemical properties and the purification technologies of proteases, as well as the methods to obtain high-yield protease strains and protease products through strain mutagenesis, genetic engineering, and other methods. For example, Lu et al. [12] screened out a high-yield, neutral protease bacterium from high-temperature yeast for making hard liquor, which was identified as *B. amyloliquefaciens*. Nie et al. [13] initially screened for microbes by calculating the transparent ring diameter, then re-screened by measuring protease activity, and finally identified two strains of high-yield proteases from Daqu in Jiang-flavor Chinese spirits, which were characterized as *B. subtilis*. However, the exact mechanism underlying the screening of high-yield proteases from Daqu of Chinese Luzhou-flavored liquor is not well understood. In the present study, we screened a strain of high-yield protease bacteria from the Daqu of ZhangGong Laojiu. We obtained the whole-genome sequence by performing WGS using the PacBio and Illumina platforms. Following gene annotations using online databases of GO, KEGG, COG, NR, and Swiss-Prot databases, we further revealed the target enriched genes and metabolic pathways. Our findings provide theoretical references for the mining of high-yield proteases and their biological functions.

## Materials and methods

### Materials and reagents

A medium-temperature starter for our study was provided by Henan ZhangGong Laojiu Wine Co., Ltd. Sodium carbonate anhydrous was obtained from Tianjin Dingshengxin Chemical Co., Ltd; the reagents L-tyrosine, agar powder, casein, trichloroacetic acid, and sodium

chloride were purchased from Tianjin Kemiou Chemical Reagent Co., Ltd; beef extract peptone was obtained from Beijing Aoboxing Biotechnology Co., Ltd; the Folin-Ciocalteu reagent was from Solarbio, Biotechnology Co., Ltd; the Ezup Column Bacteria Genomic DNA Purification Kit was from Sangon Biotech Co., Ltd.; DNA 1000 assay kit was from Lithuania Agilent Technologies; and the ABI StepOnePlus Real-Time PCR System was purchased from Life Technologies.

## Instruments

A constant-temperature incubator was purchased from Shanghai Shcimo Medical Device Manufacturing Co., Ltd; a high-speed refrigerated centrifuge was provided by Shanghai Anting Device Manufacturing Co., Ltd; a Dk-8D digital thermostatic laboratory water bath was from Guangzhou Hezhong Biotechnology Co., Ltd; a three-dimensional autoclave was obtained from Shanghai Shenan Medical Device Manufacturing Co., Ltd; 752 UV-VIS spectrophotometer was from Shanghai Jinghua Instruments; the biological purification table was from Suzhou Purification Equipment Co., Ltd; PacBio Single Molecule, real-time (SMRT) sequencing system was acquired from Pacific Biosciences of California; the Novaseq 6000 system was from Illumina; a 2100 Bioanalyzer Instrument was from Agilent Technologies; the Eppendorf 5427R centrifuge was from Eppendorf; the NanoDrop 2000 spectrophotometer was obtained from Thermal Fisher Scientific; the DYY-6C agarose gel electrophoresis kit was from Beijing Liuyi Biotechnology Co., Ltd; and finally, the PCR instrument was from Dongsheng Xingye Scientific Instrument Co., Ltd.

## Culture media

The Casein medium [14] contained: 3 g beef extract, 5 g sodium chloride, 10 g peptone, 20 g agar powder, and 4 g casein at the constant volume of 1000 mL of distilled water with a pH level between 7.0 and 7.2. The medium was sterilized at 121˚C for 20 minutes.

The beef extract peptone agar medium contained: 3 g beef extract, 5 g sodium chloride, 10 g peptone, 20 g agar powder, and 4 g casein in a constant volume of 1000 mL of distilled water at pH 7.0–7.2. The medium was sterilized at 121˚C for 20 min.

The fluid medium [15] contained: 3 g beef extract, 5 g sodium chloride, and 10 g peptone in a constant volume of 1000 mL of distilled water at pH 7.0–7.2, and sterilized at 121˚C for 20 min.

Solid medium [16]: 20 g bran and 20 mL distilled water, and sterilized at 121˚C for 20 min.

## Experimental methods

### Isolation of protease-producing bacteria.

I. *Isolation*: Daqu was ground into powder, 10 g of which was added to an Erlenmeyer flask containing 90 mL of sterile distilled water. After gently shaking the solution in a shaking incubator at 150 rpm for 30 min [17], 1 mL of the suspension was transferred to a tube containing 9 mL of sterile water and diluted 10 times. 0.1 mL of $10^{-3}$, $10^{-4}$, $10^{-5}$, and $10^{-6}$-times diluted bacteria solution was plated and cultured at 37˚C for 24 h in inverted position [18].

II. *Initial screening*: Under aseptic conditions, a sterile pipette tip was used to place a small amount of purified bacteria on the casein medium plate, created three parallels, and cultured for 24 h in inverted position to observe the formation of a transparent ring. The ratio (D/d) of transparent ring diameter (D) to cell colony diameter (d) was calculated, and those with a larger D/d ratio were selected for re-screening [19].

## Screening of protease-producing bacteria

I. *Preparation of the standard curve*: The standard curve was plotted using the absorbance value and the concentration of L-tyrosine as the ordinate and abscissa, respectively.

II. *Preparation of crude enzyme solution*: To prepare the crude enzyme solution, bacteria were inoculated into solid medium of beef extract peptone, and cultured at 37˚C for 24 h in inverted position. Subsequently, a single colony with obvious morphology was picked for inoculation into the liquid medium and then cultured on a shaker at 37˚C for 24 hours. The bacterial suspension was inoculated into the solid culture medium at 10% of the inoculation volume, and incubated at 37˚C for 3 days. Solid fermentation product (10 g) was added to a 250 mL Erlenmeyer flask containing 90 mL of distilled water and incubated in a 40˚C water bath for 1 hour. Then, the mixture was stirred every 15 minutes. After centrifugation, the crude enzyme solution was obtained [20].

III. *Measurement of protease activity using the Folin-Ciocalteu method*: Under the conditions of 1 mL of protease solution, at pH 7.5 and 40˚C, the amount of protease required to hydrolyze casein to produce 1 μg L-tyrosine per minute was defined as the protease activity, calculated using the 0

$$\text{Protease activity (U/mL)} = \frac{A \times 4 \times N \times OD}{10};$$

The constant A was obtained from the standard curve and it is equal to the number of micrograms of L-tyrosine equivalent to $OD_{680} = 1$; 4 represents 1 mL of solution collected from the total volume of 4 mL for measurement; N is the dilution multiple of protease solution to be measured; 10 represents the reaction time of 10 min, and OD represents the mean optical density of the parallel sample.

## Identification of protease-producing bacteria

I. *Morphology identification*: The high-yield, protease-producing bacteria were activated on the plate, inoculated on the casein medium by the spot inoculation method, and subjected to Gram staining. The morphology was observed under a microscope [21].

II. *Molecular characterization*: Using the DNA extracted from the high-yield, protease-producing bacteria as a template, the 16S rDNA gene in the bacterial genomic DNA was amplified using the bacterial universal primers 27F (5′–AGA GTT TCA TCT GGC TCA G–3′) and 1492R (5′–GGT TAC CTT GTT ACG ACT T–3′). PCR was performed at 94˚C for 5 minutes, followed by 35 cycles of denaturation at 94˚C for 30 seconds, annealing at 55˚C 45 seconds, and extension at 72˚C for 1.5 minutes. Finally, repair and extension at 72˚C were performed for 10 minutes, and the sample was stored at 4˚C. A 50 μL PCR amplification solution was prepared, and the amplification results were detected using 1% agarose electrophoresis to observe the specific products and their relative molecular masses. The amplified products were sent to Shanghai Sangon Biotech Co., Ltd. for sequencing [22]. Spliced sequences were compared with data in the NCBI database, and the standard bacteria with similar genetic relationships were analyzed using the MEGA 6.0, software to construct a phylogenetic tree used for visual observation of the genera and species of the bacteria [23].

*Physiological and biochemical tests*: Physiological and biochemical tests of the high-yield protease-producing bacteria were performed based on *the 8th edition of Bergey's Manual l of Determinative Bacteriology* and *the Common Identification Methods of General Bacteria* [24].

## WGS analysis

I. *DNA extraction*: Genomic DNA was extracted according to the instructions of the bacterial DNA extraction kit. To ensure the quality of WGS as well as the purity and concentration of the DNA, the samples were sent to Guangzhou Gene Denovo Biotechnology Co., Ltd. for sequencing.

II. *PacBio sequencing*: g-TUBE for genomic DNA shearing was performed to process genomic DNA into 8–10 kbp fragments, followed by DNA end repair. The obtained SMRT Bell DNA template was sent off for quality detection, and an Agilent 2100 bioanalyzer was used to evaluate the insert size. Sequencing was performed using PacBio platform.

III. *Illumina sequencing*: Genomic DNA fragments were prepared using dsDNA fragmentase, followed by end repair, phosphorylation, addition of polyA tails, and sequencing adapters. DNA fragments were purified using AMPure XP magnetic beads, and target fragments of 300–400 bp were subjected to PCR to establish the sequence library. PCR products were further purified using AMPure XP magnetic beads and examined using an Agilent 2100 bioanalyzer. Qualified products were sequenced using the Illumina Noveseq 6000 platform.

IV. *Bioinformatic analyses*: Low-quality data in the raw data (reads) of PacBio and Illumina sequencing were filtered out of the dataset to ensure the accuracy and reliability of the bioinformatic analyses. Clean data were assembled to analyze genomes containing target strains, including genomic components of strains, tandem repeat prediction, tRNA prediction, rRNA prediction, sRNA prediction, genomic island prediction, transposons prediction, and prephage prediction using TRF (version 4.09) [25], tRNAscan (version 1.3.1) [26], rRNAmmer (version 1.2) [27], cmscan (version 1.1.2) [28], IslandPath-DIMOB (version 1.0.0) [29], TransposonPSI (version: 20100822) [30], and PHAST (version 2.0) [31], respectively. Gene function annotation was performed using the following online databases: COG database (https://www.ncbi.nlm.nih.gov/COG/), KEGG database (https://www.genome.jp/kegg/), NR database (ftp://ftp.ncbi.nih.gov:21/blast/db/FASTA/), SwissProt database (https://www.uniprot.org/), and GO database (http://geneontology.org/).

## Results

### Isolation of protease-producing bacteria

I. *Isolation*: Through gradient dilution and plate coating, 296 strains of protease-producing bacteria were screened out from Daqu of ZhangGong Laojiu.

II. *Initial screening*: A total of 296 strains of protease-producing bacteria were inoculated on casein medium and cultured at 37°C for 24 h in inverted position. The calculated D/d values are listed in Table 1.

**Table 1. Diameter of transparent ring and strain diameter ratio.**

| No. | d/cm | D/cm | D/d |
|---|---|---|---|
| DW-2 | 0.60±0.06 | 2.30±0.10 | 3.84±0.25 |
| DW-7 | 0.57±0.15 | 2.50±0.06 | 4.39±0.98 |
| DW-8 | 0.50±0.06 | 1.90±0.06 | 3.80±0.34 |
| DW-22 | 0.60±0.06 | 2.20±0.12 | 3.69±0.62 |
| DW-32 | 0.60±0.06 | 2.40±0.10 | 4.00±0.25 |
| DW-40 | 0.50±0.00 | 1.80±0.06 | 3.60±0.12 |

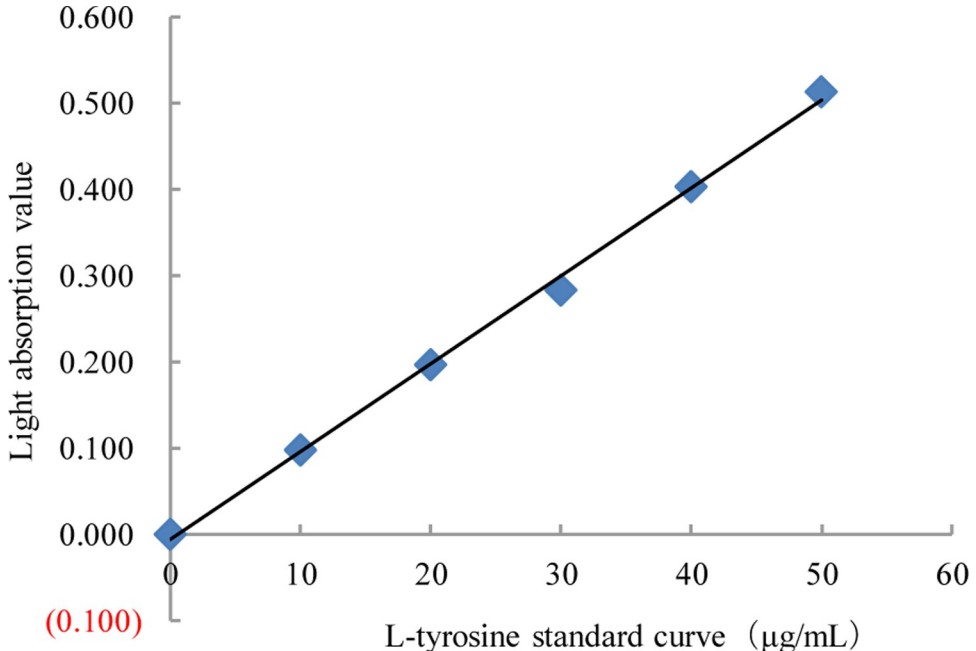

**Fig 1. L-tyrosine standard curve.**

The D/d ratio of DW-7 was 4.39 and the highest among all of the isolated strains, which is likely attributed to the strong protease activity and degradation ability of casein. Protease activity in the subsequent highest six strains was measured to detect high-yield protease-producing bacteria.

## Screening of high-yield protease-producing bacteria

As shown in Fig 1, the regression equation of the L-tyrosine standard curve was $y = 0.0102x - 0.0057$, and the correlation coefficient (R) was 0.9977, indicating a significant linear relationship between protease activity and L-tyrsoine production.

*I. Measurement of protease activity using the Folin-Ciocalteu method*

As shown in Table 2, the protease activity of DW-7 was the highest among the six strains tested at 99.54 U/mL. We subsequently examined the morphology, molecular biology, and physiological and biochemical properties of DW-7.

*II. Identification of high-yield protease-producing bacteria*

Cell colonies were milky white with a moist and viscous surface and wrinkles. They were round with smooth edges but lackED flagella (Fig 2A). Gram staining was performed to

**Table 2. Determination of protease activity.**

| No. | OD-value | protease activity (U/mL) |
|---|---|---|
| DW-2 | 0.555±0.01 | 87.56±1.79 |
| DW-7 | 0.631±0.01 | 99.54±2.23 |
| DW-8 | 0.540±0.00 | 85.19±0.11 |
| DW-22 | 0.525±0.00 | 82.82±0.00 |
| DW-32 | 0.574±0.01 | 90.55±1.90 |
| DW-40 | 0.497±0.03 | 78.41±4.24 |

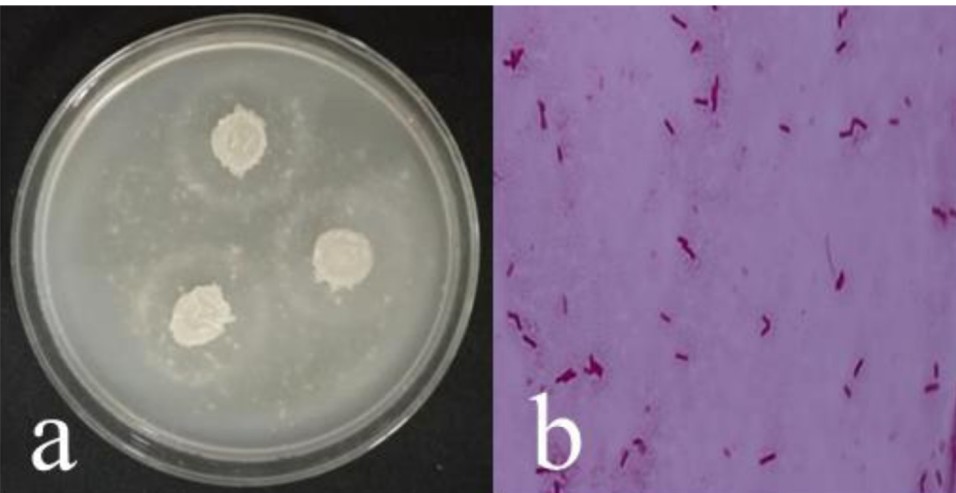

**Fig 2.** (a) Colony morphology of DW-7; (b) Morphology of DW-7 cells (magnification 1000X).

observe the cell morphology under a microscope. The bacteria were stained red, suggesting the presence of Gram-negative bacteria. The individual bacterial cells were rod-shaped (Fig 2B).

## Molecular characterization

The DNA fragment of DW-7 was amplified by PCR and visualized by agarose gel electrophoresis. A phylogenetic tree was constructed using the MEGA 6.0. As shown in Fig 3, the size of the amplified product was approximately 1,500 bp, which is consistent with that of the target product.

Comparative analysis using the NCBI database showed that DW-7 shared the closest relationship with *Bacillus velezensis*, at 97.81% similarity (Fig 4).

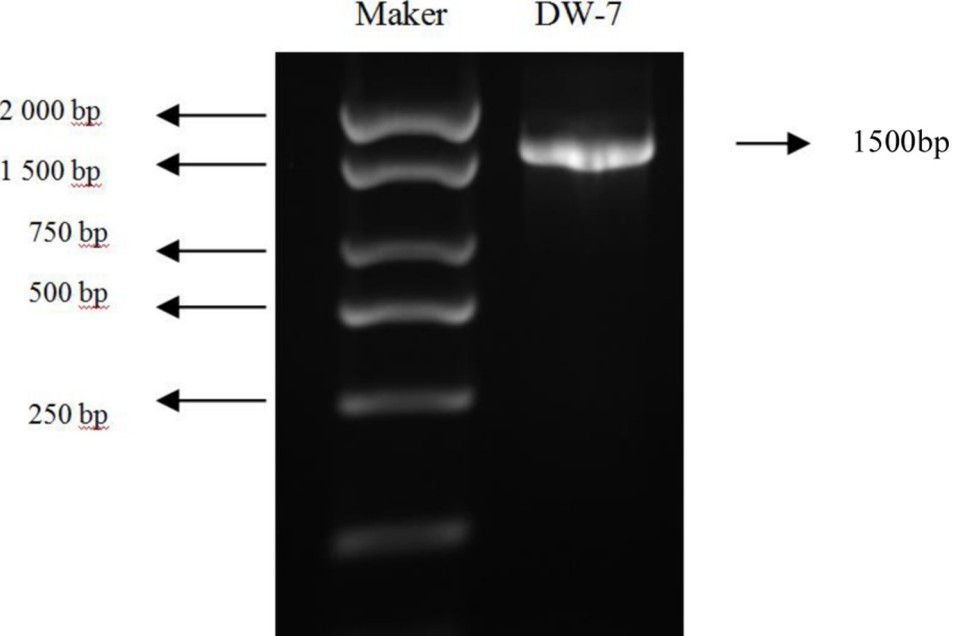

**Fig 3. Electrophoresis results of DW-7.**

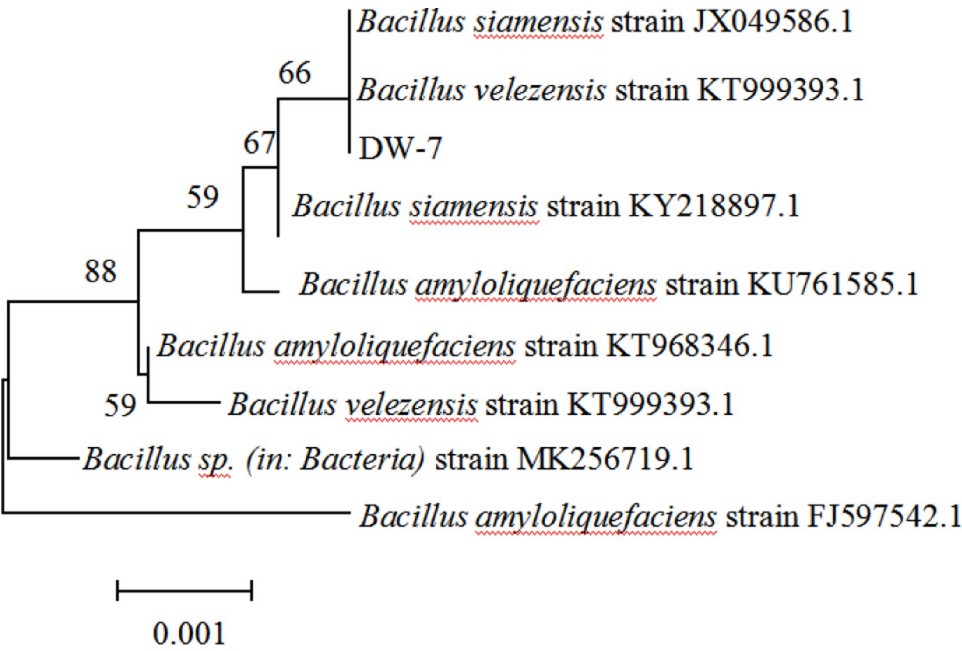

**Fig 4. Phylogenetic tree of the DW-7 bacteria.**

## Physiological and biochemical test results

DW-7 presented a positive catalase test, meaning it was able to use D-mannitol to produce acid and survive on a medium containing 10% NaCl or at pH 5.0. Citrate was not used, but propionate was available (Table 3). Based on the physiological and biochemical test results, as well as 16S rDNA sequence analysis, DW-7 was confirmed to be *Bacillus velezensis*.

## Genome assembly of DW-7

WGS of DW-7 was performed using the PacBio sequencing platform, and the results showed that there were 193,753 effective reads after filtration, with a mean sequence length of 8380.3 bp (Table 4).

The DW-7 was comprised of a circular genome of 3,942,829 bp length. Protein-encoding genes were predicted using the NCBI database. A total of 3,662 genes were identified, with a combined length of 3,402,822 bp. The longest gene length was 16,299 bp, and the shortest was 75 bp. The GC content accounted for 46.46%. There were 64 transposons in the full genome of DW-7, with a total length of 4,515 bp. Additionally, there were 24 long interspersed repeats

**Table 3. Physiological and biochemical analysis of the DW-7 bacterial strain.**

| Experiment | Result |
|---|---|
| Tolerance to 10% NaCl | + |
| Tolerance to pH 5.0 | + |
| D-mannitol acid production | + |
| Catalase test | + |
| Propionate | + |
| Citrate | - |

+ indicates a positive result, and − indicates a negative result.

**Table 4. Genome statistics of the DW-7 strain.**

| Type | Length | Amount |
|---|---|---|
| Sequence | 3942829 bp | 1 |
| Gene | 3402822 bp | 3662 |
| GC content | 46.46% | |
| Total transposon length | 4515 bp | 64 |
| Long scattered sequence | 1869 bp | 24 |
| Short scattered sequence | 1474 bp | 22 |
| Long terminal repeat sequence | 185 bp | 3 |
| tandem repeat sequences | 10086 bp | 147 |
| tRNA | 6639 bp | 86 |
| rRNA | 41367 bp | 37 |
| sRNA | 2227 bp | 17 |
| Gene Islands | 28970 bp | 1 |
| prophage | 90882 bp | 1 |

(LINEs) with a total length of 1,869 bp accounting for 0.05% of the genome, and 22 short interspersed repeats (SINEs) with a total length of 1,474 bp accounting for 0.04% of the genome. There were also three long terminal repeats (LTRs) with a combined length of 185 bp and 147 tandem repeats with a combined length of 10,068 bp, that together accounted for 0.26% of the genome. There were 86 tRNAs detected in the genome of DW-7, with a combined length of 6,639 bp and a mean length of 77 bp, as well as nine 5S rRNAs with a total length of 1,044 bp and a mean length of 116 bp. Nine 16S rRNAs with a combined length of 13,948 bp and a mean length of 1,549 bp were detected and nine 23S rRNAs with a combined length of 26,375 bp and a mean length of 2,930 bp were also detected. A total of 17 sRNAs with a combined length of 2,227 bp and a mean length of 131 bp were also detected. Finally, one genomic island was detected in the DW-7 with a total length of 28,970 bp as well as one prephage with a length of 90,882 bp.

## Bacterial genome components

Circular genome maps were depicted based on assembled genome sequences and prediction results of protein-encoding genes, which provided a visual of the genomic characteristics (Fig 5).

The outermost circle of the circular genome map of DW-7 is 3,402,822 bp in length. The second and third circles are positive and negative-strand coding regions, respectively. Different colors represent different functions annotated by the COG database. A total of 3,662 protein-encoding genes were identified, of which 2,796 (76.35%) genes were annotated by COG. Metabolism, information storage and processing, cell function and signal transmission, and atypical features were the four main functional categories. A total of 25 types were annotated, including general function prediction only, amino acid transport and metabolism, unknown function, transcription, carbohydrate transport, metabolism, and several other functions The fourth circle iscomprised of ncRNAs, in which tRNAs and rRNAs are colored in black and red, respectively. The fifth circle is the GC content, in which orange and blue indicate that the GC content was higher or lower than the mean value of the full genome, respectively. The sixth circle represents the GC-Skew value. Collectively, the circular genome maps of DW-7 directly showed the distribution of the DW-7 genome.

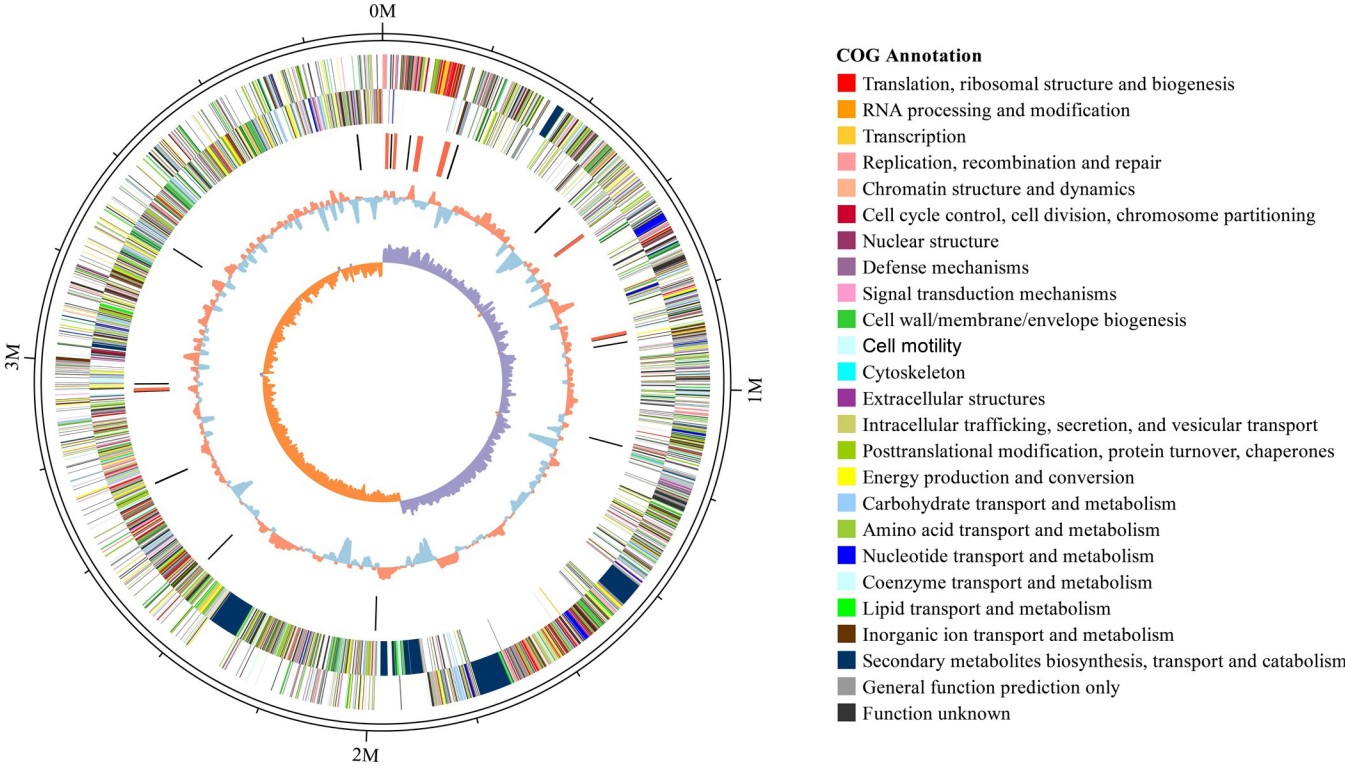

**Fig 5. Complete map of the genome of DW-7 bacteria.**

## Gene annotation using COG

After comparison of the function annotations between the DIAMOND and COG databases, annotations were classified and depicted in Fig 6.

Function annotation of identified protein-encoding genes in DW-7 was performed using COG, and a total of 2,796 protein-coding genes involving 25 categories were annotated. Among them, 445 (15.92%) protein-encoding genes were enriched in the general function prediction only, which was the most enriched category, followed by 336 (12.02%) protein-encoding genes enriched in amino acid transport and metabolism category. Furthermore, 274 (9.80%), 245 (8.76%), and 205 (7.33%) protein-encoding genes were enriched in the transcription, carbohydrate transport and metabolism, and inorganic ion transport and metabolism categories, respectively. A combined total of 302 (10.80%) protein-encoding genes were enriched in all functional positions, which necessitates further exploration.

## Gene annotation using KEGG

The amino acid sequence of DW-7 was compared with data obtained from the KEGG database, and annotation results were obtained by matching the genes of target species and their corresponding function annotation information. A total of 2,249 genes in DW-7 were enriched in 23 pathways and five major functions of metabolism, including genetic information processing, environmental information processing, cell process, and biological system (Fig 7).

In the category of metabolic processes, 12 pathways were annotated, and 242 and 206 genes were enriched in the carbohydrate metabolism and amino acid metabolism pathways, respectively. Four pathways in the of genetic information processing category were annotated, with

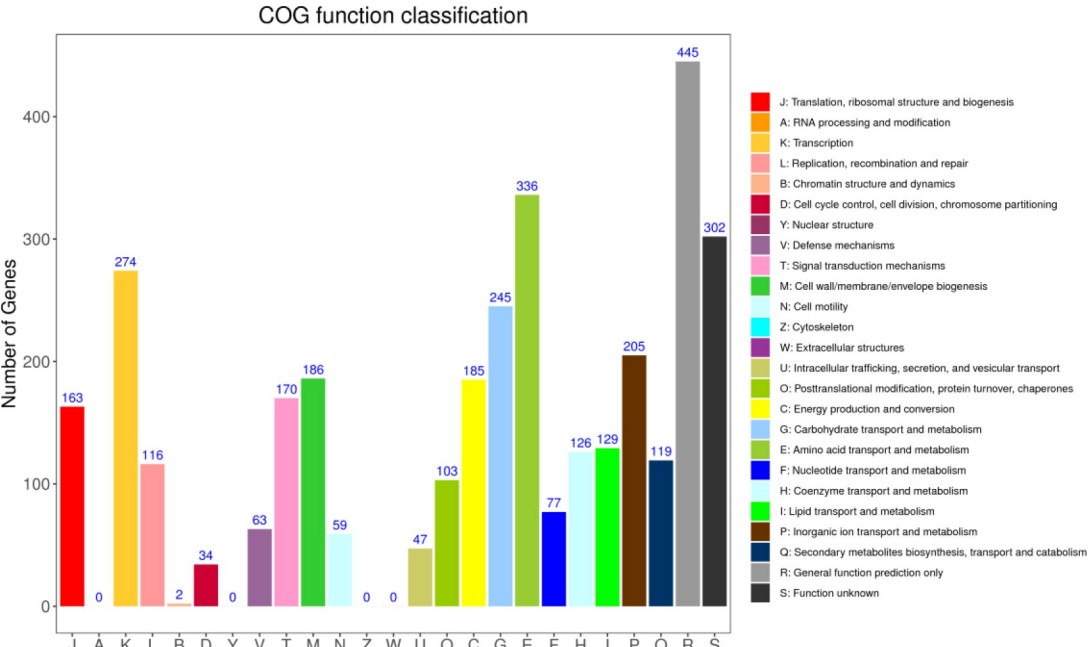

**Fig 6. Functional classification of COG.**

the translation pathway having the most annotations, in one of which 85 genes were enriched. There were 147 and 124 genes enriched in the membrane transport and signal transduction pathways, respectively, both of which are subcategories of the environmental information processing category. In addition, three pathways were annotated in the cell process category, in which the prokaryotic cell community was the most frequently annotated, in one of which 85 genes were enriched. In the biological systems category, four and three protein-encoding genes were enriched in the environmental adaptation and immune system subcategories, respectively.

According to the KEGG annotation information, 196 protein-encoding genes in DW-7 were enriched in the following pathways, all of which were correlated with presence of proteases involved in the metabolism of alanine, aspartic acid, glutamate, glycine, serine, threonine, ABC transporter, and transporter pathways. These protein-encoding genes included the KSG66-11125 (serine dehydratase), KSG66-13075 (aspartate kinase), KSG66-14990 (serine kinase), KSG66–10425 (aspartate aminotransferase), and several others.

## Gene annotation using nr and Swiss-Prot

The gene sequence of DW-7 was translated into the corresponding amino acid sequence, then compared with data from the nr database within the NCBI database (Fig 8).

A total of 3,662 genes in DW-7 were annotated using the nr database. Among them, 2,709; 389; 198; 107; 70; 53; and 41 genes were enriched in the primary seven identified species *B. velezensis*, *B. amyloliquefaciens*, *B. subtilis*, *B. abscessus*, and *Bacillus sp.5B6*, and *Streptococcus pneumoniae*, respectively.

A total of 3,399 genes in DW-7 were functionally annotated in the Swiss Prot database, which is a selected protein sequence database that describes protein function, structure, post-translational modification, mutations, and other characteristics.

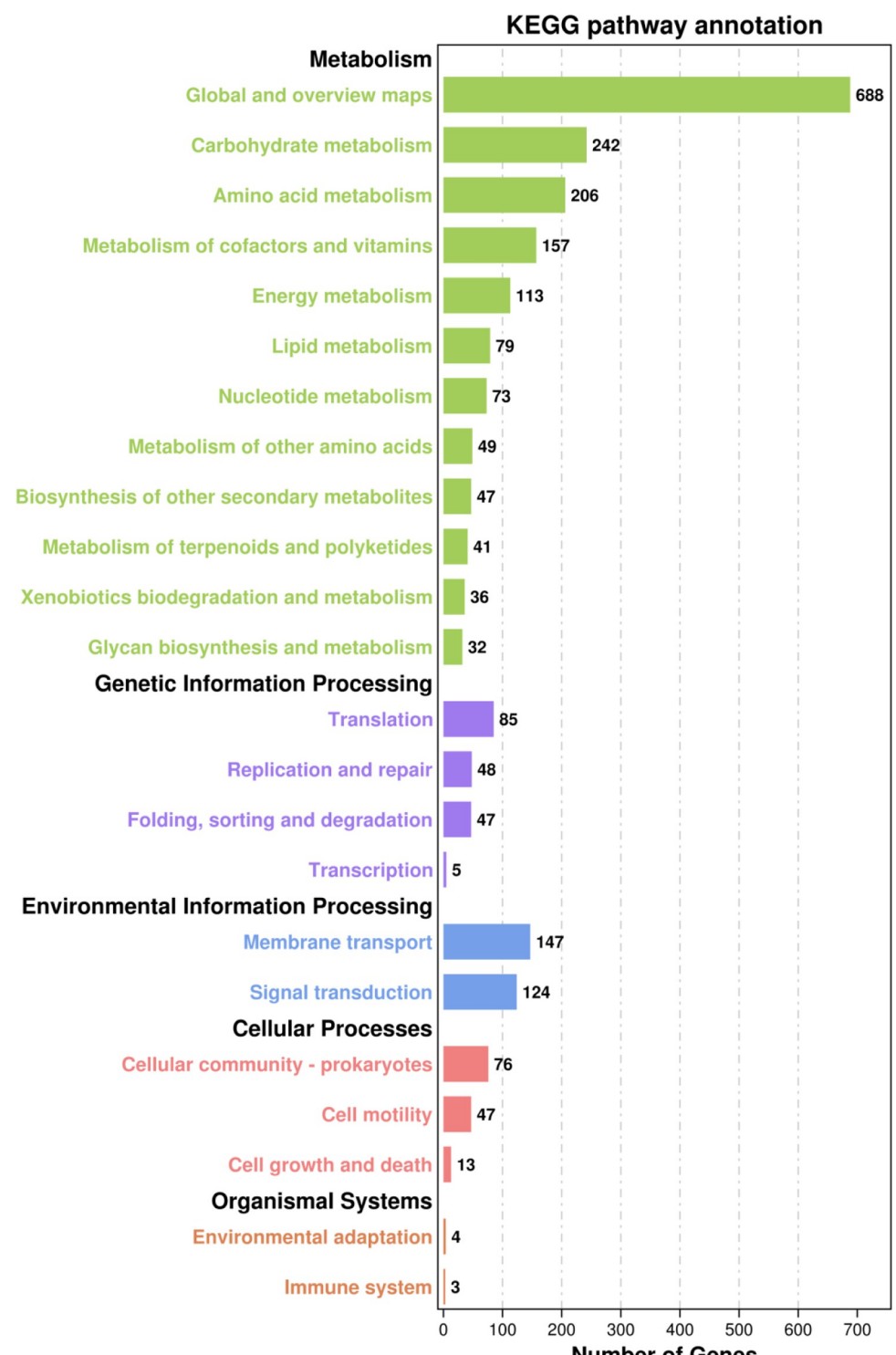

**Fig 7. Bacterial gene functional annotation KEGG metabolic pathway.**

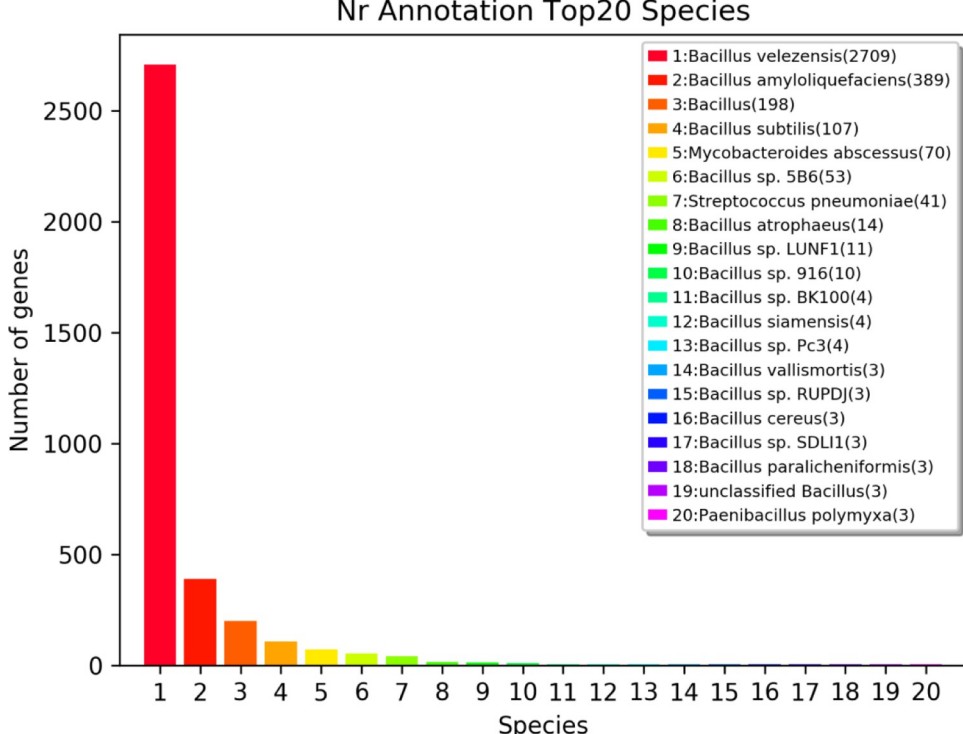

**Fig 8. Annotated species statistics of Nr database (top 20 species).**

## Gene annotation using GO

According to the annotation information in the nr database, GO function annotations were identified to comprehensively describe the attributes of genes and gene products in DW-7 (Fig 9).

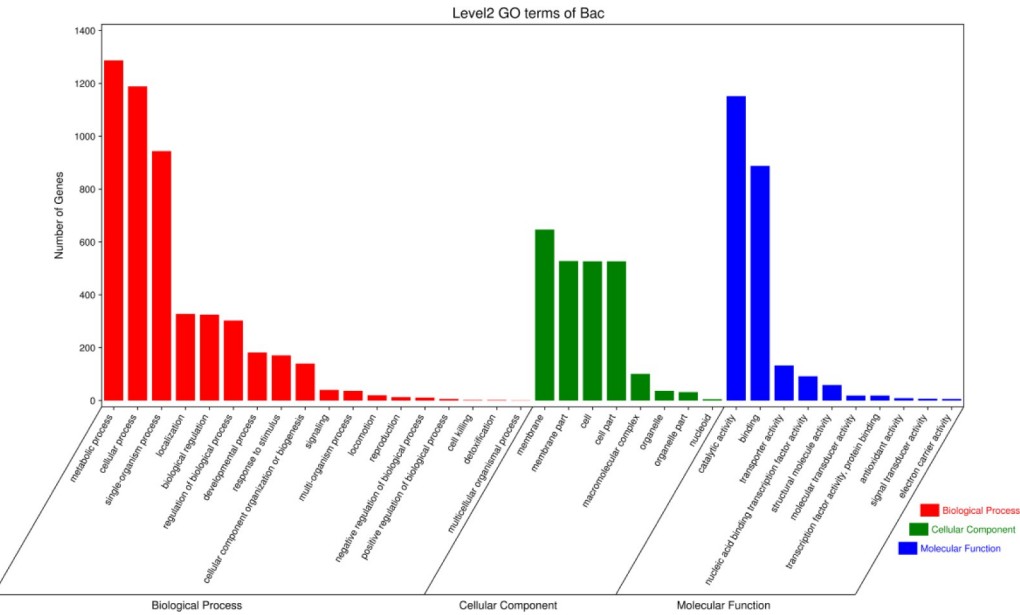

**Fig 9. GO classification of bacterial gene function annotation.**

A total of 2,127 genes from DW-7 were annotated in the GO database. GO terms including biological process (BP), cellular component (CC), and molecular function (MF), comprising 18, 10, and 10 elements, respectively. The top three annotated BPs were the metabolic processes, biological processes, and single-cell tissue processes, in which 1,287, 1,189, and 944 genes were enriched, respectively. There were 647, 528, 527, and 527 genes enriched in the top four CCs, including the cell membranes, cell membrane compositions, cells, and genes related to cell components, respectively. A total of 1,152, 888, and 133 genes were enriched in the most annotated MFs of catalytic activity term: gene and binding, and transport activity, respectively.

## Discussion

In this study, 296 strains of protease-producing bacteria were isolated from Daqu from Zhang-Gong Laojiu Wine Co., Ltd. Additionally, one strain of high protease-producing bacteria, DW-7, with an enzyme activity of 99.54 U/mL was isolated by combining the transparent circle screening method and the forinol screening method. Morphological observations, 16S rDNA molecular biology identification, as well as physiological and biochemical tests, were performed on the isolated DW-7 which was identified as *Bacillus velezensis*. Few studies have examined high protease-producing strains using baijiu Daqu as samples. Yuan Xianling et al. [14] isolated and purified five strains of protease-producing bacteria from strong baijiu Daqu, including *Bacillus thuringiensis*, *Bacillus subtilis*, *Bacillus cereus*, *Bacillus multilocularis* and *Bacillus subtilis* with enzyme activities of 41.75 U/mL, 31.90 U/mL, 53.40 U/mL, 46.10 U/mL and 44.55 U/mL, respectively; Zhao Qunli [32] et al. used Moutai-flavored baijiu Daqu as the raw material, from which they screened a strain 3 J-1 with high enzyme activity and a protease activity of 70.15 U/mL; Huang et al. [33] also screened a strain from Moutai-flavored baijiu Daqu, identified as *A. hennebergii*, with a maximum enzyme activity of 71.13 U/mL after optimization of pH, carbon source, nitrogen source and metal ions; and a high enzyme activity level of 99.54 U/mL. Protease-producing bacteria are widely found in marine [34], animal intestine [35], soil [36], and other environments, and screening for these high protease-producing strains can be useful for industrial production and food processing.

Proteases are a class of enzymes that catalyze the hydrolysis of proteins into peptides and amino acids and degrade proteins into raw materials to form small peptides or amino acids used in yeast and lactic acid bacteria fermentation as a nitrogen source to promote the growth of brewing microorganisms [37]. Some of the amino acids in the protein degradation products are flavoring substances themselves, while others are precursors of aroma components that ultimately affect the formation and quality of the flavor of white wine [38]; and thus are indispensable enzymes for white wine production. Studies have shown that the hydrolytic enzyme system in soybean currants is mainly protease [39]. When the protease content is normal, it can significantly inhibit the production of hetero-ethanol oil [40], and the protease activity of superior grade barley is higher than that of normal grade barley [41]. Deng et al. compared different colors of high-temperature barley and found that QW protease activity was higher than QR, QY, and QB, while the windiness of bacterial enzyme-encoding genes was higher in QW [42]. Numerous studies have shown that bacteria of the genus Bacillus have a strong protease production capacity [2, 43].

To further investigate the protease metabolism mechanism and metabolic pathway, we performed WGS of DW-7, and determined that the genome length of this strain was 3,942,829 bp, with a GC content of 46.45%. The combined length of DW-7's coding genes was 3,402,822 bp, and by gene function annotation, there were 2,796 genes annotated in the COG database, with the main focus on general function prediction, amino acid transport and metabolism,

transcription, carbohydrate transport and metabolism, and inorganic ion transport and metabolism. In the KEGG database, 2,283 genes were annotated, mainly in carbohydrate metabolism, amino acid metabolic pathway, translation pathway, membrane transport, and signal transduction pathway. 2,127 genes were annotated in the GO database, mainly in metabolic function, cellular process, cell membrane, cell membrane components, catalytic activity, gene, and binding. Protease genes do not have biological functions after transcription and translation to form proteins, rather only after modification by a series of chemical reactions, such as ubiquitination, phosphorylation, acetylation, glycosylation, methylation, or lipidylation, can there be more protein types with complex structures, precise regulation, specific action, and perfect function [44]. In this study, COG function prediction, KEGG metabolic pathway annotation, and GO annotation of the genes encoded by this strain showed normal gene transcription and translation and a relatively high number of annotated genes, which indicated that the enzyme metabolism of this bacterium is active. Protease gene expression metabolic pathways were mainly focused on alanine, aspartate, glutamate, glycine, serine, and threonine, as well as ABC transporter proteins; and transporter protein. The main proteases involved in these pathways were serine dehydratase, aspartate kinase, serine kinase, and aspartate aminotransferase. Prajapati et al. [45] resolved the protease gene of *Bacillus amyloliquefaciens* strain *KCP2* by WGS, while Li et al. [46] sequenced the genome of *Laceyella sacchari* FBKL4 and found that the genome contains genes encoding key enzymes, such as proteases and peptidases, along the tetramethylpyrazine metabolic pathway. Proteases are classified into four categories based on their catalytic role in a variety of physiological pathways, including serine proteases, aspartate proteases, cysteine proteases, and metalloproteases [47]. Among the proteases, serine proteases are the most common, accounting for nearly one-third of known proteases [48, 49]. The 196 protease-related gene annotations in this study were significantly enriched for serine dehydratase, aspartate kinase, serine kinase, and aspartate aminotransferase, which laterally reflected that the bacterium was active in serine protease and aspartate protease synthesis. This result also indicated that this high protease-producing *Bacillus* sp. metabolized mainly these two types of catalytic proteases.

## Conclusion

In this experiment, a strain of high-yield protease bacteria DW-7 with 99.54 U/mL protease activity was isolated from medium-temperature Daqu provided by ZhangGong LaoJiu Wine Co., Ltd. Through morphological observation, 16S rDNA sequence analysis, and physiological and biochemical tests, DW-7 was determined to be *Bacillus velezensis*. WGS analysis of DW-7 revealed 196 genes that were highly related to proteases predominantly enriched in the metabolism of alanine, aspartic acid, glutamate, glycine, serine, threonine, ABC transporter proteins, and transporter pathways.

Overall, our study preliminarily explored the biological characteristics and metabolic functions of the high-yield protease-producing strain DW-7 by WGS analysis. Abundant genomic information was obtained that provided useful references for identifying the functional characteristics of the strain, as well as the foundation for elucidating the background of the high-yield protease-producing feature. Our results are of great significance for the development of novel high-yield proteases from Daqu.

## Supporting information

**S1 File.**
(PDF)

**S1 Raw data.**
(ZIP)

**S1 Raw images.**
(PDF)

# Acknowledgments

We thank ZhangGong LaoJiu Wine Co. Ltd. for providing the samples and the distillery staff for their help in sampling. We also thank the members of the subject group for supporting this study and contributing to the experimental work. We would like to thank Editage (www. editage.cn) for English language editing.

# Author Contributions

**Conceptualization:** Zhijun Zhao.

**Writing – original draft:** Yanbo Liu, Junying Fu, Linlin Wang.

**Writing – review & editing:** Yanbo Liu, Zhijun Zhao, Huihui Wang, Suna Han, Xiyu Sun, Chunmei Pan.

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
