## [Decision Letter · Decision Letter 0]

26 Nov 2021

PONE-D-21-33082“Isolation, identification and whole genome sequencing of high-yield protease bacteria from daqu of Zhanggong Laojiu”PLOS ONE

Dear Dr. Liu,

Thank you for submitting your manuscript to PLOS ONE. After careful consideration, we feel that it has merit but does not fully meet PLOS ONE’s publication criteria as it currently stands. Therefore, we invite you to submit a revised version of the manuscript that addresses the points raised during the review process.

We look forward to receiving your revised manuscript.

Kind regards,

Tofazzal Islam, Ph.D.

Academic Editor

PLOS ONE

Journal Requirements:

“This work supported by the Major Science and Technology Projects of Henan Province of China(181100211400),Key Technologies Research and Development Program of Henan Province of China(202102110130) , Scientific Research Foundation for Docotors of Henan University of Animal Husbandry and Economy(2018HNUAHEDF011) and Key Subject Projects of Henan University of Animal Husbandry and Economy.”

“The authors received no specific funding for this work”

Reviewers' comments:

Reviewer's Responses to Questions

**Comments to the Author**

1. Is the manuscript technically sound, and do the data support the conclusions?

Reviewer #1: Yes

Reviewer #2: Yes

2. Has the statistical analysis been performed appropriately and rigorously? 

Reviewer #1: Yes

Reviewer #2: Yes

3. Have the authors made all data underlying the findings in their manuscript fully available?

Reviewer #1: Yes

Reviewer #2: Yes

4. Is the manuscript presented in an intelligible fashion and written in standard English?

Reviewer #1: Yes

Reviewer #2: Yes

5. Review Comments to the Author

Reviewer #1: Check the Reference format. Some references are unavailable with the title search. The findings are interesting. Enough data has been represented to support the article. Overall, this is a nice article!

Reviewer #2: The manuscript is interesting and write up is good. However, the significance of identifying, screening and whole genome sequencing of high-yield protease bacteria are not clear. The author should add some sentences to justify their research.

Some other corrections are listed below.

Line 16: Correct spacing problem as U/mL (Abstract)

Line 43: Bacillus should be Italic (Introduction, 2nd paragraph)

-No need to write Bacillus in every cases. Write the genus name first and then write only the short form like B. amyloliquefaciens, B. subtilis etc throughout the manuscript (Line 72, 75)

-Figure caption (Fig.4 or Fig 5 etc) is not similar. Please follow the exact rules of writing.

-Is it necessary to discuss what other did in their experiment in the conclusion sections? My recommendation is to transfer them in the discussion section. I surprised there is no discussion section. Please add discussion sections and discuss based on the previous findings. Highlighted the significant findings, suggestions, further research possibilities in conclusion section.

6. PLOS authors have the option to publish the peer review history of their article (what does this mean?). If published, this will include your full peer review and any attached files.

Reviewer #1: No

Reviewer #2: No

---

## [Author Response · Author response to Decision Letter 0]

12 Jan 2022

Dear Editor and Reviewers:

Thank you for your letter and for the reviewers' comments concerning our manuscript. Those comments are all valuable and very helpful for revising and improving our paper, as well as the important guiding significance to our researches. We have studied comments carefully and have made correction which we hope meet with approval. Revised portion are marked in yellow in the paper. The manuscript has been polished by Editage. The main corrections in the paper and the responds to the reviewer 's comments are as following:

Response to Reviewers 

1.Believing ‘Zhanggong Laojiu’ as the name of an alcoholic beverage and isolated from a host (yeast), the place (origin of source) could be mentioned here.

The medium-temperature Daqu（A traditional starter used to make Chinese baijiu） produced by ZhangGongLaoJiu Wine Co. Ltd.( Brands of alcoholic beverages)

2. Better to use the full form when mentioned for the first time.

Already added

3. Shorter keywords are preferable.

The keywords have been shortened and summarized

4. fusel oil?

Some of the alcohols present in alcoholic beverages, including 1-propanol, iso-butanol, 1-butanol, 2-butanol, iso-amyl alcohol, active amyl alcohol etc.

5. Was there any verification performed with protease-producing gene sequences obtained from annotation?

The sequence of the protease-producing gene has not been verified yet, which is the next step we will do.

6. Words should be in italic where required. Some references are not available on google scholar (or even on google search) upon searching with the title.

A correction has been made, some references are only published in China, and may not be found on Google search for this reason.

7. What is ‘spore spore’?

There was an error in this area and changes have been made.

We tried our best to improve the manuscript and made some changes in the manuscript. These changes will not influence the content and framework of the paper. We appreciate for Editor and Reviewers' warm work earnestly, and hope that the correction will meet with approval. If there are any shortcomings in the article, please tell me immediately, and I will seriously revise it again.Once again, thank you very much for your comments and suggestion.

Thank you and best regards.

Yours sincerely,

Yanbo Liu E-mail: yanboliu@hnuahe.edu.cn

---

## [Editor Report · Decision Letter 1]

16 Feb 2022

“Isolation, identification and whole genome sequencing of high-yield protease bacteria from daqu of Zhanggong Laojiu”

PONE-D-21-33082R1

Dear Dr. Yanbo Liu,

We’re pleased to inform you that your manuscript has been judged scientifically suitable for publication and will be formally accepted for publication once it meets all outstanding technical requirements.

Kind regards,

Tofazzal Islam, Ph.D.

Academic Editor

PLOS ONE
---

## [Editor Report · Acceptance letter]

8 Apr 2022

PONE-D-21-33082R1 

Isolation, identification, and whole-genome sequencing of high-yield protease bacteria from Daqu of ZhangGong Laojiu 

Dear Dr. Liu:

I'm pleased to inform you that your manuscript has been deemed suitable for publication in PLOS ONE. Congratulations! Your manuscript is now with our production department. 

Kind regards, 

on behalf of

Professor Dr. Tofazzal Islam 

Academic Editor

PLOS ONE